# Blockchain Technology, Enterprise Risk and Enterprise Performance

**Ye Zhen \*, Wen Qiao, Ruyuan Wang and Wenli Wang**

School of Economics and Management, Taiyuan University of Science and Technology, Taiyuan 030024, China; s202216110660@stu.tyust.edu.cn (W.Q.); 18734168576@163.com (R.W.); wlwang@tyust.edu.cn (W.W.)

\* Correspondence: zhenye009@126.com

**Abstract:** In order to explore the impact of the application of blockchain technology on enterprise performance, as well as the mechanism of enterprise risk and the information disclosure quality on this impact process, and taking the data of A-share listed companies in China's manufacturing industry from 2015 to 2022 as a research sample, this paper adopts methods such as multi-period difference-in-differences (DID) modeling to conduct an empirical investigation. Findings: The application of blockchain technology can improve enterprise performance. Enterprise risk plays a partial mediating effect, because blockchain technology can reduce enterprise risk and thereby improve enterprise performance. Information disclosure quality has an inhibitory influence on the process by which blockchain technology affects enterprise risk and a facilitating influence on the process by which enterprise risk affects enterprise performance. The results show that manufacturing enterprises with low information disclosure quality can reduce enterprise risk by combining with blockchain technology in production, management, and other aspects, thus improving enterprise performance and promoting sustainable development of enterprise economy.

**Keywords:** blockchain technology; enterprise risk; enterprise performance; information disclosure quality





## 1. Introduction

Blockchain is a new application model of computer technologies such as distributed data storage, peer-to-peer transmission, consensus mechanisms, encryption algorithms, and more. First proposed by Satoshi Nakamoto in 2008, blockchain instigated a new round of technological and industrial changes in the world as soon as it was introduced. The technology records and validates transaction data in a decentralized manner, giving all participating members the right to monitor and share immutable records in the network [1]. It also reshapes enterprise business models, organizational forms, and business models in terms of transparency, accountability, trust, security, efficiency, and cost minimization [2–4]. At present, it has already been widely applied in industries such as finance, Internet of Things (IoT), public services, digital copyright, insurance, real estate, law, logistics supply chain, and navigation industries [5,6]. Blockchain improves the traceability and transparency of the supply chain by recording data from each logistics node onto the blockchain, thus enabling real-time monitoring and verification of information such as transport, quality, and temperature of goods [7]. It can also offer the supply chain system greater credibility, transparency, efficiency, and security, and at the same time improve the collaborative efficiency of enterprises, optimize the relevant processes, and promote resource sharing and cooperation among enterprises, thus promoting the modernization of the industrial chain and supply chain [8,9]. At the 18th Collective Study of the Political Bureau of China's Central Committee in October 2019, President Xi Jinping emphasized the need to rapidly promote the development of blockchain technology and industrial innovation, thereby building a blockchain industrial ecosystem. Today, all industries

are actively exploring the application scenarios of blockchain; Alibaba, Tencent, JD.com, and other Internet industry giants are increasing their investment in the research and development of blockchain to seize market opportunities.

Manufacturing directly reflects the productivity level of a country and is an important factor that distinguishes developing and developed countries. In recent years, blockchain technology has played a greater role in solving problems arising in manufacturing production, logistics, quality, and after-sales service. For example, Walmart has applied blockchain technology to its supply chain traceability system, effectively solving the problem of counterfeiting in its supply chain and allowing customers to buy genuine products. Blockchain technology can also effectively prevent the risk of information leakage and malicious manipulation brought about by malicious attacks and control of any single-node device in the Industrial IoT, as well as allow employees to grasp the operational conditions of all devices in real time. Therefore, many scholars have explored the impact of blockchain applied to manufacturing enterprises, pointing out that blockchain can not only improve the production process and reduce operating costs, but also improve enterprise profitability and optimize the internal operation of the organization [10–12]. The application of blockchain technology in manufacturing enterprises enables enterprises to establish a new production organization and optimize workflow, thus providing an effective solution for the intelligent transformation of enterprises [13]. Blockchain can improve the performance of collaborative innovation in enterprises, reduce the complexity of inter-team collaboration, increase data trustworthiness, enhance transparency and mutual trust in the process of collaborative innovation, and promote a smarter and more automated style of conducting business, which has significant implications for corporate governance [14]. Blockchain is an innovative technology that promises to change the decision-making process and can provide innovative organizational practices for the development of digitalization in enterprises [15], which can not only improve supply chain management outside the enterprise but also have an impact on the internal operational aspects of the enterprise.

Some scholars have also studied the relationship between blockchain and enterprise performance as well as the relationship between blockchain and financial risk, pointing out that when enterprises combine blockchain technology in production, management and other aspects, their credit and reliability have been significantly improved. From the perspective of production and operation, blockchain technology can improve the efficiency of firm operations [16]. The impact of blockchain applications on enterprise performance depends on the degree of correlation between the blockchain technology implementation link and the main business of the enterprise. The higher the degree of correlation, the better the positive promotion of the enterprise performance. The lower the degree of association, the worse the positive promotion of the enterprise performance, or a negative impact can even arise [17]. From the perspective of corporate governance, Lin and Wu (2021) [18] suggested that blockchain technology better promotes enterprise performance when the corporate governance structure exhibits a higher proportion of independent directors and higher ownership concentration. From the perspective of sustainable performance, Di Vaio and Varriale (2020) [19] stated that the main impact of blockchain technology on operations management in supply chain management is that it facilitates cooperation between crucial players to reduce fragmented, inefficient, and uncoordinated operations. Lu, Ru, and Han (2022) [20] argued that the combination of blockchain and financial risk can help the management to clearly grasp the internal operation of the enterprise and then make the right decision to reduce the financial risk of the enterprise. Xu, Chen, and Wang (2022) [21] studied the impact of diversification on the level of enterprise risk-taking under blockchain enablement. Chen, Pan, and Qi (2022) [22] explored the theoretical feasibility of applying blockchain to enterprise risk management.

In summary, it can be seen that although there has been literature focusing on the impact of blockchain technology on business performance, it has also explored the relationship between blockchain technology and financial risk at the level of blockchain technology architecture and system application. However, the path of the relationship that arises be-

tween blockchain technology, enterprise risk, and enterprise performance need to be tested. Improving the information disclosure quality is an intrinsic requirement of promoting the sustainable development of listed companies, and it is also conducive to improving the value of the company and increasing corporate performance [23]. Adequate information disclosure is the best path to achieve a reasonable response to corporate performance [24]. Therefore, whether the information disclosure quality will have an impact on the path of blockchain technology to enhance firm performance is yet to be empirically investigated. This study deeply explores the mechanism of the impact of blockchain technology application on the performance in manufacturing firms through a large sample of empirical analyses from the perspective of information disclosure, which enriches the research on blockchain technology and its effects from the perspective of information disclosure. We first use listed companies in the manufacturing industry as the research object to empirically investigate the impact of blockchain technology on the enterprise performance and explore the mediating effect played by enterprise risk in the process. Then, we further analyze the impact of the information disclosure quality on the relationship among blockchain technology, enterprise risk, and enterprise performance as well as the path of blockchain technology to improve enterprise performance.

## 2. Theoretical Analysis and Research Hypotheses

### 2.1. Blockchain Technology and Enterprise Performance

Blockchain technology builds a tamper-proof data recording mechanism and a smart contract, automated-execution transaction mechanism, which improves the processing efficiency of data and information while ensuring data security, optimizing personnel allocation and resource dispatch, and thus reducing the management cost of the enterprise.

Specifically, the application of distributed ledgers promotes information sharing among enterprises, enabling them to search for the required information resources in real time without the need to use a third-party platform and therefore effectively reduce the cost of information searching for enterprises. The consensus mechanism, under the premise of ensuring the authenticity of the data, solves the problem of collaboration and trust in transactions that do not require mutual trust as a condition, which not only improves the enterprise's trust system but also reduces the cost of trust between enterprises. Smart contracts have the functions of automatic execution and real-time monitoring, which can avoid human intervention in the transaction process and monitor the flow of funds in real time [25], thus reducing the post-transaction costs and supervision costs of enterprises to a certain extent. The application of blockchain technology can reduce the cost of enterprises from different perspectives, improve the enterprise performance, and then achieve the economic sustainability of enterprises. In summary, this paper proposes the following Hypothesis 1:

**Hypothesis 1 (H1).** *Applying blockchain technology can improve enterprise performance.*

### 2.2. The Impact of Enterprise Risk on the Relationship between Blockchain Technology and Enterprise Performance

The application of blockchain technology can create a highly trusted trading environment for enterprises [26]. It effectively reduces the probability of enterprise risks and ensures the chances of sustainable operation. On the one hand, the blockchain can automatically update the enterprise financial accounts in real time. Every piece of changed financial information forms a copy in the blockchain, so as to avoid the behavior of financial forgery within the enterprise and reduce the risk of financial fraud, which helps to protect the normal production and operation of the enterprise. On the other hand, the open and transparent mechanism of blockchain can avoid collusion between enterprises. Multiple enterprises can collaborate and supervise each other, which not only reduces the risk of default but also helps to promote the formation of a deep and long-term cooperation mechanism between enterprises to further reduce the risk of enterprise operation and then truly

realize the improvement of the enterprise performance [27]. In summary, the following Hypothesis 2 is proposed:

**Hypothesis 2 (H2).** *Enterprise risk plays a mediating effect in the impact of blockchain technology on enterprise performance.*

*2.3. Information Disclosure Quality Moderates the Mediating Role of Enterprise Risk between Blockchain Technology and Enterprise Performance*

Information disclosure quality refers to the degree of accuracy, comprehensiveness, timeliness, understandability, and comparability of the amount of information conveyed by an enterprise when disclosing information to the public. According to Asymmetric Information Theory and Signaling Theory, low-quality information disclosure increases the information asymmetry between external investors and firms, while high-quality information disclosure enhances investor confidence, which reduces investors' prediction of corporate risk [28]. Enterprises with high information disclosure quality generally have better profitability, growth, and risk-control ability. Therefore, the effect of adopting blockchain technology to reduce enterprise risk for such enterprises is not obvious. For enterprises with low information disclosure quality, the introduction of blockchain technology can not only make each process transparent but also enable enterprises to optimize each process individually according to the problems existing in each process, which in turn reduces enterprise risk. The following Hypothesis 3a is proposed:

**Hypothesis 3a (H3a).** *Information disclosure quality has an inhibitory influence on the process by which blockchain technology affects enterprise risk.*

Based on Signaling Theory, firms with low information disclosure quality generally have poorer profitability and are reluctant to disclose too much internal information about their firms, and even tend to disclose false information [29]. This results in the asymmetry of enterprise risk information between external and internal enterprises. These information asymmetries will magnify the effect of enterprise risk, cause investors to panic about risk, and lead to a decline in enterprise performance. For enterprises with high information disclosure quality, the difference between external and internal enterprise risk information is small, which will reduce the effect of enterprise risk, mitigate investors' panic about risk, and lead to an increase in enterprise performance. Hypothesis 3b is thus proposed:

**Hypothesis 3b (H3b).** *Information disclosure quality has a facilitating influence on the process by which enterprise risk affects enterprise performance.*

In summary, the theoretical model of this paper is shown in Figure 1.

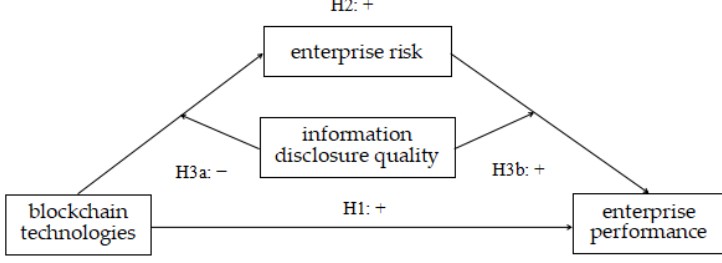

**Figure 1.** Theoretical model.

## 3. Research Design

*3.1. Sample Selection and Data Source*

This paper selects manufacturing companies listed in China's A-share market from 2015 to 2022 as the object of the initial study. This study uses Baidu to search for core

reports such as the 2022 China Top 100 Industrial Blockchain Enterprises Ranking and the Circular of the General Office of the Ministry of Industry and Information Technology on the Announcement of the List of Typical Blockchain Application Cases in 2022 to obtain an initial list of enterprises applying blockchain technology to solve the company's practical problems [30,31]. The relevant data for each enterprise comes from the Database of Chinese Industrial Enterprises by the National Bureau of Statistics of China, CSMAR database, China manufacturing industry reports, annual reports of various companies, and related news. The samples were screened as follows: First, firms listed after 2015 are excluded. Second, ST and *ST firms are excluded. ST and *ST stocks indicate that the company's financial or other conditions are abnormal. The exclusion of ST and *ST makes the sample more representative of the market in general. Third, firms with missing values for the variables are excluded. Eventually, a total of 1049 sample firms was obtained, with the firms adopting blockchain technology as the treatment group and the remaining as the control group. These samples had 8392 observations, which included aspects such as the period of adoption of blockchain technology, the enterprise risk metrics, the degree of information disclosure quality, a range of enterprise financial indicators, and the structural characteristics of the enterprise.

Currently, the application of blockchain technology in the field of supply chain finance is relatively rich, and the supply chain finance enterprise mostly occurs in the manufacturing industry. Therefore, this paper selects listed companies in the manufacturing industry as the object of study with certain rationality. The total number of companies adopting blockchain technology is 45 out of a sample of 1049, and the adoption of blockchain technology is spread over the period 2017–2020. Considering that the subsequent tests need to analyze the changes in enterprise performance during the period before and after the implementation of blockchain technology for all the enterprises that have implemented blockchain technology, the data range of the sample 2015–2022 therefore needs to include the time range 2017–2020 for the adoption of blockchain technology by the treatment group. Therefore, the data were selected at a reasonable time. Subsequently, the data were analyzed using stata15.1 software.

### 3.2. Variable Selection

Dependent variable: Tobin's Q reflects the present and future value of a company, fully reflects the growth of a company, objectively demonstrates the economic sustainability of the enterprise, and is difficult for managers to manipulate. The larger the Tobin's Q, the higher the enterprise performance; conversely, the smaller the Tobin's Q, the lower the enterprise performance. Therefore, Tobin's Q was used to measure the enterprise performance [32].

Independent variable: Blockchain technology is used as the independent variable, measured by the virtual variable Blockchain. The virtual variable Blockchain is taken as 1 if the sample has adopted blockchain technology in that year; conversely, the virtual variable Blockchain is taken as 0 if the sample has not adopted blockchain technology in that year.

Mediating variable: Enterprise risk, as the mediating variable in this paper, is measured with the Z-score model for listed manufacturing firms, with the symbol Z denoting the calculation result [33]. Z is calculated by a number of financial indicators; the result is not only objective but also can fully reflect the enterprise risk, so the choice of Z value to measure enterprise risk has a certain degree of rationality. The larger the Z, the smaller the enterprise risk; the smaller the Z, the larger the enterprise risk.

Moderating variable: Information disclosure quality is measured with the KV index [33]. KV index indicates the degree of influence of trading volume information on the rate of return. The principle of the KV method is as follows: when the degree of influence of trading volume information on the rate of return is high, investors rely more on the reference of trading volume information rather than the disclosure information as the basis of investment, which indicates that the information disclosure quality of enterprises is low;

on the contrary, when the degree of influence of trading volume information on the rate of return is small, it indicates that the information disclosure quality of enterprises is high.

Control variables: In constructing the regression model, this paper selects control variables from financial indicators and organizational structure [16–18]. Specifically, these include liquidity ratio, asset–liability ratio, total asset turnover, cost and expense margin, the growth rate of total operating income, the structure of governance, the structure of shareholding, and the age of the company. In order to minimize the regression error, the age of the company is taken as a logarithmic number. The variable symbols, names, and meanings are shown in Table 1.

**Table 1.** Definition of the main variables.

| Variable Type | Variable Name | Variable Symbol | Variable Definition | | |
|---|---|---|---|---|---|
| Dependent variable | Enterprise performance | Q | Expressed in terms of Tobin's Q-value Tobin's Q = market capitalization/total assets | | |
| Independent variable | Blockchain technology | Blockchain (treat × post) | Firm i adopts blockchain technology in year t, taking the value of 1, otherwise. | treat | Group virtual variable |
| | | | | post | Time virtual variable |
| Mediating variable | Enterprise Risks | Z | Computed by the Altman Z-score model (1968) [33] $Z = 1.2X_1 + 1.4X_2 + 3.3X_3 + 0.6X_4 + 0.999X_5$ | | |
| Moderating variable | Information disclosure quality | KV | Referring to the approach of Xu et al. (2015) [34] $KV = \ln\left\lvert \frac{P_t - P_{t-1}}{P_{t-1}} \right\rvert = \alpha + \beta\left(\frac{Vol_t}{Vol_0} - 1\right) + \mu_i$ | | |
| Control variables | Current ratio | CR | liquid asset/liquid liability × 100% | | |
| | Asset–liability ratio | Lev | Total liabilities/Total assets × 100% | | |
| | Total asset turnover | Turnover | Total sales revenue/average total assets × 100% | | |
| | Ratio of profits to cost | RPCE | Total profit/total cost × 100% | | |
| | Gross operating income growth rate | Growth | (Total operating income for the current period − total operating income for the base period)/total operating income for the base period × 100% | | |
| | Governance structure | Gov | Number of independent directors/ numbers of Director | | |
| | Shareholding structure | Top1 | Percentage of shares held by the largest shareholder | | |
| | Age of the company | Age | Logarithmic values for observation year − IPO year values | | |

### 3.3. Model Building

We construct a multi-period DID model of the impact of blockchain technology on enterprise performance [35]. The independent variables in the model can be represented by the virtual variable $Blockchain_{it}$.

$$Q_{it} = \alpha_0 + cBlockchain_{it} + \beta_1 Control_{it} + \delta_{it} + \varepsilon_{it} \tag{1}$$

To further examine whether blockchain technology has an impact on enterprise performance through the mediating effect of enterprise risk according to H2, the stepwise method is used to test the mediating effect of firm risk [36]. The following mediating effect model is constructed on the basis of model (1):

$$Z_{it} = \alpha_0 + aBlockchain_{it} + \beta_1 Control_{it} + \delta_{it} + \varepsilon_{it} \tag{2}$$

$$Q_{it} = \alpha_0 + c\prime Blockchain_{it} + bZ_{it} + \beta_1 Control_{it} + \delta_{it} + \varepsilon_{it} \tag{3}$$

We model the moderating effect of information disclosure quality in the process of blockchain technology affecting enterprise risk:

$$Z_{it} = \alpha_0 + \beta_1 Blockchain_{it} + \beta_2 KV_{it} + \beta_3 Control_{it} + \delta_{it} + \varepsilon_{it} \tag{4}$$

$$Z_{it} = \alpha_0 + \beta_1 Blockchain_{it} + \beta_2 KV_{it} + \beta_3 Blockchain_{it} \times KV_{it} + \beta_4 Control_{it} + \delta_{it} + \varepsilon_{it} \tag{5}$$

We model the moderating effect of information disclosure quality in the process of enterprise risk affecting enterprise performance:

$$Q_{it} = \alpha_0 + \beta_1 KV_{it} + \beta_2 Z_{it} + \beta_3 Control_{it} + \delta_{it} + \varepsilon_{it} \tag{6}$$

$$Q_{it} = \alpha_0 + \beta_1 KV_{it} + \beta_2 Z_{it} + \beta_3 KV_{it} \times Z_{it} + \beta_4 Control_{it} + \delta_{it} + \varepsilon_{it} \tag{7}$$

In models (1)–(5), subscript *i* denotes enterprise and subscript *t* denotes year. The independent variable is the *Blockchain_{it}*, which takes 1 if the firm *i* adopts blockchain technology in year *t*, and 0 otherwise; *Control_{it}* is the control variable, $\delta_{it}$ is the individual fixed effect, and $\varepsilon_{it}$ is the random perturbation term.

## 4. Analysis of Empirical Results

### 4.1. Descriptive Statistics

The descriptive statistics of the two groups of samples applying blockchain technology and not applying blockchain technology in this paper are shown in Tables 2 and 3. The mean value of Q of the treatment group applying blockchain technology is 2.22, and the mean value of Q of the control group not applying blockchain technology is 1.99. The former is larger than the latter, which to some extent presents the trend that the application of blockchain technology can improve the enterprise performance, and the preliminary judgment is that Hypothesis 1 is reasonable. The mean value of Z for companies adopting blockchain technology is 6.21, greater than the mean value of Z for companies not adopting blockchain technology, which is 5.21, indicating that companies adopting blockchain technology are less risky. To some extent, this result shows the trend that adopting blockchain technology can reduce enterprise risk. The mean value of KV of firms applying blockchain technology is 0.47, less than the mean value of KV of firms not adopting blockchain technology, 0.52, which suggests that the information disclosure of firms applying blockchain technology is relatively adequate.

**Table 2.** Descriptive statistics for key variables (blockchain = 1, application of blockchain technology).

| Variable | N | Mean | SD | Min | Max |
|---|---|---|---|---|---|
| Q | 360 | 2.22 | 2.04 | 0.14 | 13.45 |
| Z | 360 | 6.21 | 9.23 | −1.65 | 90.92 |
| KV | 360 | 0.47 | 0.20 | 0.02 | 1.23 |
| CR | 360 | 3.05 | 4.93 | 0.49 | 49.63 |
| Lev | 360 | 0.38 | 0.20 | 0.03 | 0.93 |
| Turnover | 360 | 0.51 | 0.29 | 0.01 | 1.66 |
| RPCE | 360 | 0.06 | 0.31 | −3.49 | 0.66 |
| Growth | 360 | 0.12 | 0.48 | −0.67 | 5.39 |

**Table 3.** Descriptive statistics for key variables (blockchain = 0, no blockchain technology applied).

| Variable | N | Mean | SD | Min | Max |
|---|---|---|---|---|---|
| Q | 8032 | 1.99 | 1.68 | 0.10 | 25.71 |
| Z | 8032 | 5.21 | 6.69 | −5.10 | 160.85 |
| KV | 8006 | 0.52 | 0.23 | 0 | 2.03 |
| CR | 8032 | 2.26 | 2.18 | 0.11 | 42.72 |
| Lev | 8032 | 0.41 | 0.18 | 0.01 | 1.72 |
| Turnover | 8032 | 0.67 | 0.40 | 0.01 | 3.70 |
| RPCE | 8032 | 0.10 | 0.23 | −3.56 | 5.38 |
| Growth | 8032 | 0.20 | 1.62 | −0.91 | 82.70 |

*4.2. Parallel Trend Test*

The parallel trend assumption is a prerequisite for the use of the multi-period DID model. The treatment group and the control group have a common trend in the enterprise performance before adopting blockchain technology, which guarantees that the difference in enterprise performance that exists between the treatment and control groups after the implementation of the technology is a net effect caused by the blockchain technology [35]. In this paper, the following model is constructed:

$$
\begin{aligned}
Q_{it} = {} & \alpha + \beta_1 event_{it}^{-5} + \beta_2 event_{it}^{-4} + \beta_3 event_{it}^{-3} + \beta_4 event_{it}^{-2} + \beta_5 event_{it}^{0} + \beta_6 event_{it}^{1} \\
& + \beta_7 event_{it}^{2} + \beta_8 event_{it}^{3} + \beta_9 event_{it}^{4} + \beta_{10} event_{it}^{5} + \lambda Control_{it} + \delta_i + \varepsilon_{it}
\end{aligned}
\tag{8}
$$

In the above equation, when the superscript of the variable event is $n < 0$, it denotes the nth year before firm i adopts blockchain technology; $n = 0$ indicates that firm i started adopting blockchain technology in that year; $n > 0$ denotes the nth year after firm i adopts blockchain technology. The first period before the adoption of blockchain technology is used as the base period, so the parallel trend test lacks data from the $-1$ period [37]. The results are shown in Figure 2.

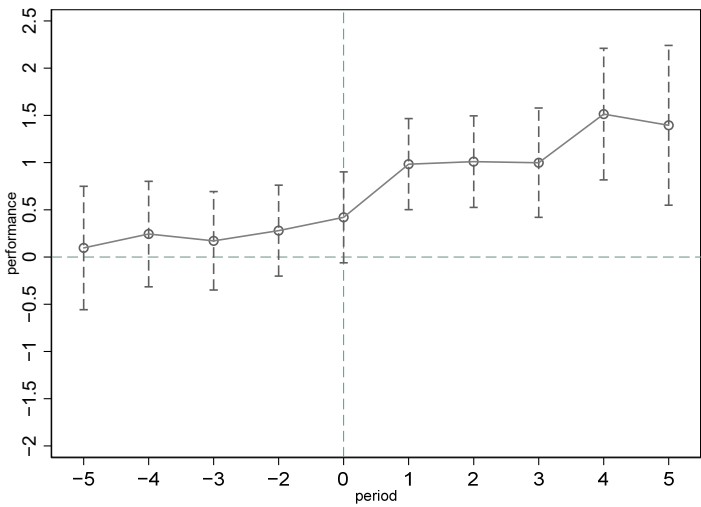

**Figure 2.** Parallel trend test.

The coefficient of the virtual variable of relative time before adopting the blockchain technology is insignificant in value, indicating no significant difference between the treatment group and the control group in terms of the enterprise performance before adopting the technology, which satisfies the parallel trend hypothesis. Further, in terms of dynamic effects, the impact of blockchain technology adoption on the enterprise performance has not yet stabilized in the two years following the adoption of the technology; two years after the adoption of the technology, the coefficient of the impact of the application of blockchain technology on the enterprise performance is significantly positive and increasing, indicating that the application of blockchain technology can produce a technological effect that promotes the improvement of the enterprise performance, but with a certain lag.

*4.3. Multiple Regression*

Before using the panel data regression analysis with 1049 sample observations for the period 2015–2022, the data were standardized in order to avoid excessive differences in the values of the variables affecting the regression results. The results of the base model regression and the mediating effect model regression are shown in Table 4, subject to the precondition that the parallel trend test is satisfied.

**Table 4.** Analysis of regression results.

| Variable | Q Model (1) | Z Model (2) | Q Model (3) |
|---|---|---|---|
| Blockchain | 0.713 *** | 2.317 *** | 0.266 ** |
| | (5.22) | (5.38) | (2.44) |
| Z | | | 0.193 *** |
| | | | (65.60) |
| Constant | 11.239 *** | 22.333 *** | 7.122 *** |
| | (32.48) | (19.56) | (25.27) |
| Control | Yes | Yes | Yes |
| Individual FE | Yes | Yes | Yes |
| N | 8392 | 8392 | 8392 |
| R-squared | 0.15 | 0.547 | 0.467 |

Note: *** indicates significance at the 1% level; ** indicates significance at the 5% level.

According to Hypothesis 1, the coefficient c represents the effect of the independent variable blockchain technology on the dependent variable the enterprise performance. The magnitude of the coefficient c indicates the difference in the enterprise performance among the samples that adopt blockchain technology and the samples that do not and is expected to be positive. The coefficient of the variable Blockchain in model (1) of Table 4 is 0.713 and is significantly positive at the 1% level, indicating that the application of blockchain technology improves the enterprise performance. Therefore, H1 is valid.

Model (2) tests the effect of the independent variable blockchain technology on the mediating variable enterprise risk. The coefficient of Blockchain is significantly positive, which indicates that the adoption of blockchain technology can improve the Z and reduce enterprise risk. Model (3) tests the combined effect of the independent variable blockchain technology and the mediating variable enterprise risk on the dependent variable of enterprise performance. The coefficient of Z in model (3) is positive at the 1% significance level, while the coefficient of Blockchain is also significantly positive, but its coefficient decreases from 0.713 to 0.266, and the significance level decreases from 5.22 to 2.44, which suggests that enterprise risk plays a mediating role in the impact of blockchain technology on the enterprise performance. Therefore, H2 is valid.

*4.4. Robustness Test*

4.4.1. PSM-DID

Considering the differences in individual characteristics between enterprises that have adopted blockchain technology and those that have not, in order to avoid the endogenous problem caused by selectivity bias, the PSM-DID method was adopted to examine the impact of blockchain technology on enterprise performance [38].

In conducting propensity score matching, the sample firms were divided into two groups: a group of "firms that adopted blockchain technology in 2017 and subsequent years" for the treatment group (treat = 1), and a group of "firms that did not adopt blockchain technology in any of the years 2015–2022" for the control group (treat = 0) [39]. Then, in this paper, using the Logit model, the nearest neighbor matching of one pair of two, one pair of three, one pair of four, and one pair of five with a caliper distance of 0.5 was performed. CR, Lev, Turnover, Gov, and Top1 from the control variables were used as matching variables.

Nearest neighbor matching of one pair of two, one pair of three, one pair of four, and one pair of five all passed the balance test. Taking one-to-two nearest neighbor matching as an example, the results of the balancing test of the matched data are shown in Table 5. It can be seen that the standardized bias of all variables after matching are all less than 10% and lower than the differences before matching. Meanwhile the *p*-values of all variables were significant before matching and non-significant after matching, indicating that there was no systematic difference between the experimental and control groups after matching, which passed the test of balance.

**Table 5.** Balancing test results.

| Variable | Unmatched Matched | Mean Treated | Control | %Bias | %Reduct \|Bias\| | t | P > \|t\| |
|---|---|---|---|---|---|---|---|
| CR | U | 3.05 | 2.26 | 20.9 | | 6.27 | 0.000 |
| | M | 2.80 | 2.79 | 0.1 | 99.6 | 0.01 | 0.991 |
| Lev | U | 0.38 | 0. 41 | −17.5 | | −3.42 | 0.001 |
| | M | 0.38 | 0.40 | −6.3 | 64.0 | −0.82 | 0.414 |
| Turnover | U | 0.51 | 0.67 | −45.1 | | −7.42 | 0.000 |
| | M | 0.51 | 0.52 | −2.3 | 94.9 | −0.37 | 0.712 |
| Gov | U | 0.39 | 0.38 | 29.6 | | 5.86 | 0.000 |
| | M | 0.40 | 0.40 | −4.3 | 85.4 | −0.52 | 0.603 |
| Top1 | U | 3.06 | 4.01 | −9.8 | | −1.64 | 0.101 |
| | M | 3.07 | 3.05 | 0.2 | 97.6 | 0.03 | 0.973 |

The changes in standardized bias before and after matching of variables are shown in Figure 3. Comparing the results before matching, the standardized bias is significantly reduced for most variables, and the propensity score matching results can be considered valid.

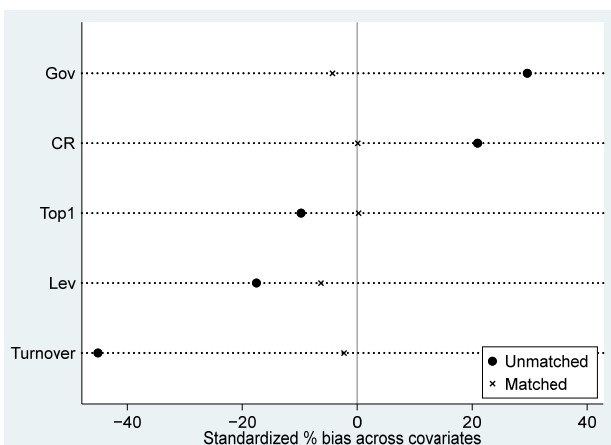

**Figure 3.** Change in standardized deviation.

To further test the quality of propensity score matching, the kernel density function was plotted, as shown in Figure 4. It can be seen that the difference between the treatment group and the control group before matching is large, while the agreement between the kernel density distributions of the treatment group and the control group after matching is substantially higher, indicating better results for propensity score matching.

The empirical results of PSM-DID on the impact of blockchain technology on the enterprise performance are shown in Table 6. Columns (1)–(4) show the regression results after one pair of two, one pair of three, one pair of four, and one pair of five nearest neighbor matching, respectively. Regardless of the matching method used, the coefficient of Blockchain is always significantly positive, indicating that the application of blockchain technology improves the enterprise performance, consistent with the results of the benchmark regression.

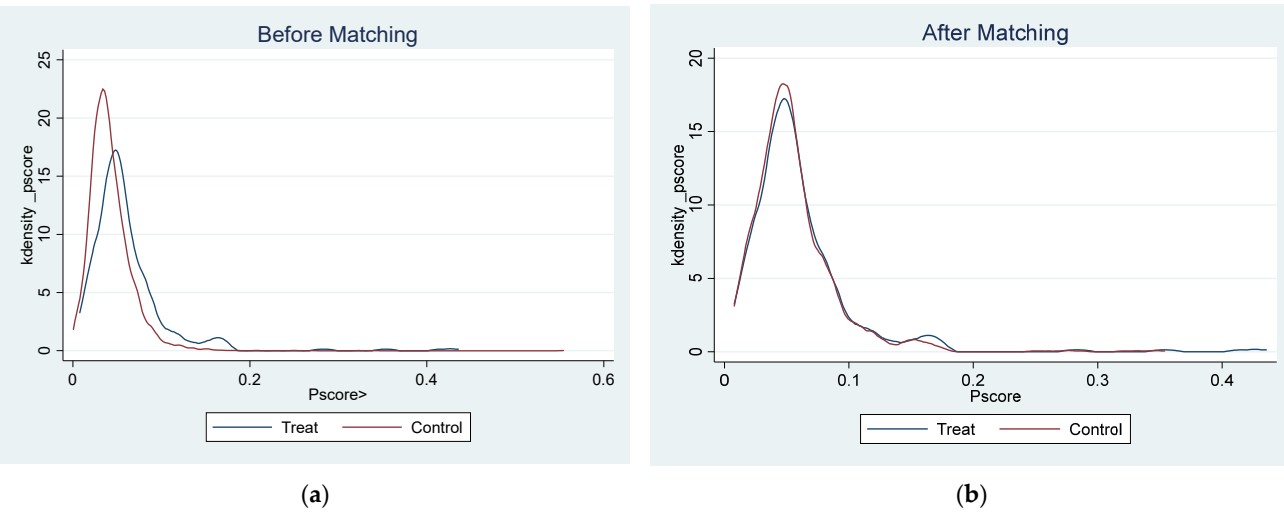

**Figure 4.** Kernel density function plot. (**a**) Before matching; (**b**) after matching.

**Table 6.** PSM-DID regression results.

| Variable | Q | | | |
|---|---|---|---|---|
| | (1) | (2) | (3) | (4) |
| Blockchain | 0.524 ** | 0.598 *** | 0.607 *** | 0.619 *** |
| | (2.26) | (2.71) | (3.00) | (3.23) |
| Constant | 9.186 *** | 10.274 *** | 10.298 *** | 10.362 *** |
| | (7.50) | (9.43) | (10.83) | (11.95) |
| Control | Yes | Yes | Yes | Yes |
| Individual FE | Yes | Yes | Yes | Yes |
| N | 1013 | 1315 | 1592 | 1869 |
| R-squared | 0.182 | 0.171 | 0.167 | 0.161 |

Note: *** indicates significance at the 1% level; ** indicates significance at the 5% level.

### 4.4.2. Placebo Testing

In order to exclude the possibility that the enhancing effect of blockchain technology on the enterprise performance receives interference from omitted variables, a placebo test for the baseline regression was taken by randomly selecting the year from the sample and randomly selecting the experimental group [40,41]. The paper was repeated with 500 random samples, and regression analyses were performed using the benchmark model; the results are shown in Figure 5. The estimated coefficients are mostly concentrated in the range of −0.8 to 0.7, and the coefficient of 0.713 estimated by the benchmark regression is a relatively obvious outlier; at the same time, the *p*-values corresponding to the estimated coefficients are overwhelmingly higher than that corresponding to the benchmark regression coefficients of 0.01, which suggests that the unobserved variables have virtually no effect on enterprise performance and that the differences in enterprise performance are due to the blockchain technology.

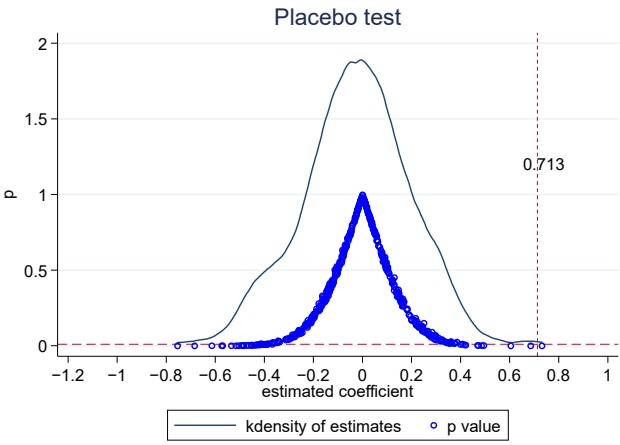

**Figure 5.** Placebo test results.

### 4.4.3. Instrumental Variables

This paper may have endogenous problems from reverse causation. This is reflected in the fact that the higher the enterprise performance, the lower the enterprise risk, which makes the firm more likely to try and embrace blockchain technology. In order to reduce the impact of this endogenous effect on the results of this paper, this study uses a two-stage least squares method to deal with the endogenous problem [42,43]. This paper selects the level of urban Internet development as an instrumental variable, denoted by City [44,45]. When the city where the enterprise is located is Hangzhou, Shenzhen, Guangzhou, Zhuhai, Xiamen, Nanjing, Shanghai, Beijing, Wuhan, and Suzhou, the variable City takes 1, otherwise it takes 0.

The instrumental variable regression results are shown in Table 7. In the first-stage regression, the coefficient of City × post is significantly positive at the 1% level, and the F-statistic is much larger than the critical value of 10; this suggests that whether a firm adopts blockchain technology is highly positively correlated with the city's level of Internet development. Specifically, the higher the level of Internet development in a city, the more likely it is that firms will adopt blockchain technology, so the condition of relevance is satisfied. In the second-stage regression, the coefficient of Blockchain is significantly positive at the 1% level. It indicates that, after considering the endogenous issue, the adoption of blockchain technology can still improve enterprise performance.

**Table 7.** Instrumental variables regression results.

| Variable | First Phase | Second Phase |
|---|---|---|
| | **Blockchain (Treat × Post)** | **Q** |
| City × post | 0.94 *** | |
| | (79.63) | |
| Blockchain (treat × post) | | 0.517 ** |
| | | (2.57) |
| Constant | −0.38 *** | 11.107 *** |
| | (−17.92) | (30.85) |
| Control | Yes | Yes |
| Individual FE | Yes | Yes |
| N | 8392 | 8392 |
| R-squared | 0.53 | |
| F-value | 904.67 | |

Note: *** indicates significance at the 1% level; ** indicates significance at the 5% level.

## 5. Expansive Research

### 5.1. Mediating Effect Test

In order to further test the accuracy of the mediating effect of enterprise risk, we used the Sobel test and Bootstrap test to verify the robustness of the mediating effect [46]. The results are shown in Table 8. The *p*-values for each test in the Sobel test was less than 0.01. The results show that the mediating effect of enterprise risk is significant. The indirect effect was 44.7%. The confidence intervals for both the indirect and direct effects of the Bootstrap test do not contain 0, indicating that a partial mediating effect of enterprise risk is established.

**Table 8.** Mediating effect test.

|  |  | Coef. | P > |Z| | Normal-Based [95% Conf. Interval] | |
|---|---|---|---|---|---|
| Sobel | Sobel | 0.44729341 | $8.213 \times 10^{-8}$ | | |
|  | Goodman-1 (Aroian) | 0.44729341 | $8.241 \times 10^{-8}$ | | |
|  | Goodman-2 | 0.44729341 | $8.185 \times 10^{-8}$ | | |
|  | Indirect effect | 0.447293 | $8.2 \times 10^{-8}$ | | |
|  | Direct effect | 0.265679 | 0.014564 | | |
|  | Total effect | 0.712972 | $1.8 \times 10^{-7}$ | | |
| Bootstrap | ind_dff | 0.4472934 | 0.000 | 0.1991548 | 0.6954321 |
|  | dir_dff | 0.2656785 | 0.032 | 0.0225232 | 0.5088339 |

### 5.2. Moderating Effect Analysis

The regression results of the moderating effect of information disclosure quality on the relationship between blockchain technology and enterprise risk are shown in Table 9. Models (4) and (5) examine the moderating effect of information disclosure quality on the relationship between blockchain technology and enterprise risk. In model (5), the interaction term of Blockchain × KV is significantly positive. Combined with the regression result of model (2), it can be seen that when KV is higher, the information disclosure quality is lower, and the application of blockchain technology will have a better effect on reducing enterprise risk; when KV is lower, the information disclosure quality is higher, and the application of blockchain technology is less effective in reducing enterprise risk. Therefore, H3a is valid.

**Table 9.** Moderating effect test.

| Variable | Z Model (4) | Z Model (5) | Q Model (6) | Q Model (7) |
|---|---|---|---|---|
| Blockchain | 2.315 *** | 0.834 | 0.271 ** | 0.279 ** |
|  | (5.38) | (1.06) | (2.50) | (2.59) |
| KV | 1.565 *** | 1.503 *** | 0.533 *** | 0.734 *** |
|  | (6.92) | (7.34) | (9.36) | (10.56) |
| Z |  |  | 0.191 *** | 0.211 *** |
|  |  |  | (64.90) | (42.56) |
| Blockchain × KV |  | 3.242 ** |  |  |
|  |  | (2.25) |  |  |
| KV × Z |  |  |  | −0.033 *** |
|  |  |  |  | (−5.02) |
| Constant | 20.782 *** | 20.75 *** | 6.995 *** | 6.764 *** |
|  | (19.02) | (19.00) | (24.91) | (23.81) |
| Control | Yes | Yes | Yes | Yes |
| Individual FE | Yes | Yes | Yes | Yes |
| N | 8366 | 8366 | 8366 | 8366 |
| R-squared | 0.551 | 0.552 | 0.474 | 0.476 |

Note: *** indicates significance at the 1% level; ** indicates significance at the 5% level. To avoid the results being affected by missing values, samples with missing KV values were excluded when performing the regression.

Models (6) and (7) in Table 9 test the moderating effect of information disclosure quality on the mediating path in the process of enterprise risk affecting the enterprise performance. The independent variables Blockchain, KV, and the mediator variable Z in model (6) are all significant, and the coefficient of the interaction term KV × Z is significant in model (7), indicating that the moderated mediating effect holds. In particular, the interaction term KV × Z is negatively related to the enterprise performance, suggesting that when the KV is higher, the information disclosure quality is lower, and the effect of reducing enterprise risk on improving enterprise performance is worse; when KV is smaller, the information disclosure quality is higher, and the effect of reducing enterprise risk on improving enterprise performance is better. Therefore, H3b is valid.

## 6. Conclusions and Apocalypse

### 6.1. Conclusions

This paper adopts the panel data of 1049 manufacturing enterprises listed on China's A-share market from 2015 to 2022 and draws the following conclusions by examining the relationship between blockchain technology, enterprise risk, enterprise performance, and information disclosure quality:

The magnitude of Q is a necessary indicator of the level of performance of the firm. From the sample data, the mean value of Q is 2.003. From the regression results, if the manufacturing company adopts blockchain technology in its business and production activities, it will increase the Q to 0.713. Compared to 2.003, 0.713 is an unusually significant value, so this paper argues that blockchain technology can significantly improve enterprise performance [17,18].

In the regression results, if the manufacturing enterprise adopts blockchain technology, the Z will increase to 2.317. The mean value of Z in the sample enterprises is 5.252, so 2.317 is a large value relative to 5.252. This leads to the conclusion that blockchain technology can improve enterprise performance by reducing enterprise risk. Therefore, enterprise risk plays a partial mediating role.

Information disclosure quality not only inhibits the process of using blockchain technology to reduce enterprise risk but also promotes the process of reducing enterprise risk to improve enterprise performance.

Manufacturing enterprises with low information disclosure quality make their production and management more transparent after adopting blockchain technology. Therefore, it reduces enterprise risk, increases the information disclosure quality, improves enterprise performance, and promotes the sustainable development of enterprise economy.

### 6.2. Apocalypse

Manufacturing firms aiming to achieve sustainable development should continue to innovate and introduce advanced scientific technology. Strengthening the research and application of advanced technology can promote the intelligent upgrade and sustainable development of enterprises. Blockchain technology can strengthen the trust mechanism between enterprises, effectively reduce transaction costs, and improve enterprise performance. Therefore, manufacturing enterprises should actively pay attention to the latest development trend of blockchain and actively introduce blockchain technology in combination with the actual situation of enterprises. Through effective integration of existing resources, they should formulate blockchain application strategies and accelerate the development of the blockchain industrial ecosystem.

Enterprises start by identifying business needs and assessing existing resources. They identify resources that can be improved through blockchain technology and those that can be directly integrated into blockchain applications. The next step is to identify key application scenarios that are closely related to the strategic objectives and the core business. Given the complexity of blockchain technology, they should consider finding partners and professional technical support to develop a detailed implementation plan, including time-

lines, resource requirements, and budget. Finally, they implement blockchain technology, monitor critical metrics and results, and make timely adjustments to strategies and plans.

Managers of a company must constantly focus on the risk threshold of the company while maximizing the company's profits. Enterprises should use blockchain technology to identify the risk points in time according to their own situation and control the risk within a reasonable range. Possible risks should be prevented and fully controlled, according to which the enterprise's risk response strategy should be adjusted.

The Global Reporting Initiative (GRI) is a standard developed to improve the transparency and quality of corporate sustainability reporting. Enterprises can refer to the GRI to consciously fulfill their information disclosure obligations, innovate information dis-closure methods, and strengthen the breadth and depth of information disclosure. After identifying disclosure priorities, enterprises should publicly explain the measures taken in information gathering and quality control. The most important and valuable information is clearly communicated to shareholders, customers, and other stakeholders through channels such as the enterprise's website, annual report, social media, and various events.

We examine the impact of blockchain technology on risk reduction and performance enhancement, mainly based on the economic and technological sustainability of enterprises. However, the consideration for other aspects of corporate sustainability is lacking, especially the impact of environmental factors on development. More importantly, it remains to be examined through which type of risk control the impact of blockchain technology on business performance is created.

**Author Contributions:** Conceptualization, R.W. and Y.Z.; methodology, Y.Z.; software, W.Q.; validation, W.Q. and R.W.; formal analysis, W.Q.; investigation, R.W.; resources, W.W.; data curation, W.Q.; writing—original draft preparation, R.W.; writing—review and editing, W.Q.; visualization, Y.Z.; supervision, Y.Z. All authors have read and agreed to the published version of the manuscript.

**Funding:** National Natural Science Foundation of China: Research on Supply Chain Finance Risk Transmission Mechanism and Prevention and Control Strategies under Major Emergencies (Project No. 72171162); Ministry of Education Youth Fund for Humanities and Social Sciences Research: Research on Digital Copyright Pledge Financing Mode and Its Risk Management Mechanism Based on Blockchain (Project No. 21YJC630172); Ministry of Education Youth Fund for Humanities and Social Sciences Research: Research on Innovative Mechanisms of Supply Chain Financing for Order Farming Under the Support of Commercial Insurance (Project No. 20YJC630148); Shanxi Philosophy and Social Science Planning Project (Think Tank Special Project): Countermeasures and Suggestions for Increasing the Supply of Quality Cultural Products and Services in Shanxi Province (Project No. W20231003); Shanxi Higher Education Institutions Philosophy and Social Science Research Project: The Impact of Environmental Regulation and Technological Innovation on the Transformation and Upgrading of Shanxi's Manufacturing Industry under the Goal of "Dual-Carbon Research (Project No. 2021W082).

**Informed Consent Statement:** Informed consent was obtained from all subjects involved in the study.

**Data Availability Statement:** All data, models, and code generated or used during the study appear in the submitted article.

**Conflicts of Interest:** The authors declare no conflict of interest.

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
