# Peer review of "Blockchain Technology, Enterprise Risk and Enterprise Performance"

_sustainability, doi:10.3390/su16010070_

Round 1

Reviewer 1 Report

Comments and Suggestions for Authors

Dear Authors,

I have reviewed the manuscript “Blockchain Technology, Enterprise Risk, and Enterprise Performance” Manuscript ID: sustainability-2684403 that has been submitted for publication in the: Sustainability (ISSN 2071-1050), and I have identified a series of aspects that in my opinion must be addressed in order to bring a benefit to the manuscript.

The article under review will be improved if the authors address the following aspects in the text of the manuscript:

1.     The abstract uses some complex language and sentence structures, which can make it challenging for readers who are not familiar with the topic to understand the research objectives and findings. It should be written in a more straightforward and reader-friendly manner.

2.     Length: The abstract is quite long and contains a lot of information for an abstract. Typically, abstracts are concise summaries of research, and this one could be trimmed down to include only the most essential points.

3.     Absence of Data Results: While the abstract mentions that the results show that the application of blockchain technology can improve enterprise performance and reduce enterprise risk, it doesn't provide any quantitative data or statistics to support these claims.

4.     Your model or research becomes more robust when tested against various datasets.

5.     Comparing your research with other researchers is a common practice in academic and scientific work. Such comparisons can help you position your research within the existing body of knowledge, identify gaps, and demonstrate the novelty or significance of your work.

6.     The references need to be updated for the years 2021 and 2022, as this field has been recently raised.

 https://doi.org/10.3390/healthcare10061110

10.21608/DUSJ.2022.275552

7.     The authors should provide more details in the conclusion about the future work that will be done later.

Reviewer 2 Report

Comments and Suggestions for Authors

The article presents a current topic and the intention of researching the use of blockchain in China's manufacturing industry is important. The article presents an adequate quantitative analysis and the results justify the hypotheses. I suggest, to improve the article in the introduction, detail and explain how you intend to investigate the impact of blockchain technology on the enterprise performance, improve the theoretical foundation of blockchain with the addition of more articles and books and detail the conclusion.

Reviewer 3 Report

Comments and Suggestions for Authors

Before your paper is ready to be published in the Sustainability journal, some improvements should be made.

1.       Consider breaking down the introduction into smaller, more focused paragraphs. This will make it easier to read and understand. For example, you can have separate paragraphs for the significance of blockchain, its applications in manufacturing, academic findings, and the research gap.

2.       Please check your citation format! The current format is totally wrong!

3.       While you mention various applications of blockchain in manufacturing, you could briefly describe how blockchain is applied in each of these scenarios.

4.       You briefly mention that some academics have pointed out the risks of corporate enthusiasm for blockchain technology, but it would be beneficial to clearly state the research gap or question that your study aims to address. What specific aspects of enterprise performance and risk are you investigating, and what are your research objectives?

5.       Where is your research significance? What are the potential practical implications of your findings for manufacturing companies in China or elsewhere?

6.       It's essential to be transparent about how the data was collected. Specify which specific datasets and sources you used. Please write down the link of the data sources or at least cite them in your references! If you used specific reports, databases, or publications, consider citing them to give credit to the original sources and provide additional context.

7.       Explain your rationale for selecting firms listed after 2015, excluding ST and *ST firms, and firms with missing values. This information will help readers understand the robustness of your sample selection process.

8.       You mentioned that you obtained 1,050 samples and 7,350 observations, which is valuable information. However, it would be beneficial to clarify what these observations represent and how they relate to the sample.

9.       Explicitly connect the sample selection and data sources to your research objectives. How does this data help address the questions or hypotheses you outlined in your introduction?

10.   In the conclusion part, if possible, include quantitative data or statistics from your study to reinforce the findings. Mention specific performance improvements, risk reductions, and the extent of the mediating effect.

11.   When making statements, especially those related to the first and second findings, provide references or examples from your data or the literature. This adds credibility and context to your conclusions.

12.   The explanation of the moderating effect of information disclosure quality is a bit complex. You might consider providing a simplified, concrete example or scenario to illustrate how it works in practice.

13.   Explicitly mention the practical implications of your findings for manufacturing companies. How can they apply blockchain technology to improve performance and reduce risk? What should they consider in terms of information disclosure quality?

14.   Suggest possible directions for future research. What other aspects of blockchain technology, risk, and performance could be explored in subsequent studies?

15.   The apocalypse part in your paper is complicated

16.   Offer more concrete advice for manufacturing enterprises. For instance, how can they effectively integrate existing resources and formulate blockchain application strategies? Providing a step-by-step approach or practical steps would be valuable.

17.   Provide more specific guidance on how manufacturing enterprises can improve their information disclosure quality. What are the best practices in terms of disclosure methods and content? Are there industry standards they should follow?

Comments on the Quality of English Language

Ensure the text is free from grammatical errors and typos.

Reviewer 4 Report

Comments and Suggestions for Authors

This article uses a large number of statistical methods to analyze and test the data. The content of the research is quite good. Please make the following suggestions for correction or explanation:

1.   Assumptions 1 and 2 are not marked in Figure 1.

2.  There are very few research documents on this topic in the Web of Science database, only 5, and there are not many references in this article. How to explain the importance of this research?

3.   This article selects manufacturing companies listed on China's A-share market from 2015 to 2021 as the preliminary research object, and uses Baidu to search for core reports such as "China's Top 100 Industrial Blockchain Companies in 2022". I would like to ask why the data does not include 2022 ?

4.    The manufacturing companies involved in this article first adopted blockchain technology in 2017. Why did the data selection start in 2015?

5.   “This paper selects the first 5 years of data of the enterprises that are the latest to adopt blockchain technology to the last 4 years of data of the enterprises that are the earliest to adopt blockchain technology.” Please explain the connection to questions 3 and 4 and the actual data range.

6.   When constructing the regression model, this article selects control variables from multiple perspectives. What is the basis for the selection? Is there any literature support?

7.  "H1 is established, which is consistent with the research results of Deng[8] and Lin[9]." Why is it necessary to repeat the study?

Comments on the Quality of English Language

The English language needs moderate editing and grammatical correction.

Round 2

Reviewer 3 Report

Comments and Suggestions for Authors

The authors' manuscript has undergone a thorough review process, and I am pleased to recommend its acceptance for publication in the Sustainability journal.

Comments on the Quality of English Language

Some parts still need improvement.